# Isolation and Characterization of Germline Stem Cells in Protogynous Hermaphroditic *Monopterus albus*

**DOI:** 10.3390/ijms23115861

**Published:** 2022-05-24

**Authors:** Xiaoyun Sun, Binbin Tao, Yongxin Wang, Wei Hu, Yuhua Sun

**Affiliations:** 1Institute of Hydrobiology, Chinese Academy of Sciences, Wuhan 430072, China; sunxiaoyun@ihb.ac.cn (X.S.); taobinbin@ihb.ac.cn (B.T.); wangyongxin@ihb.ac.cn (Y.W.); 2University of Chinese Academy of Sciences, Beijing 100049, China; huwei@ihb.ac.cn; 3The Innovation Academy of Seed Design, Chinese Academy of Sciences, Wuhan 430072, China

**Keywords:** the swamp eel, germline stem cells, ovary, testis, ovotestis, transplantation

## Abstract

Germline stem cells (GSCs) are a group of unique adult stem cells in gonads that act as important transmitters for genetic information. Donor GSCs have been used to produce offspring by transplantation in fisheries. In this study, we successfully isolated and enriched GSCs from the ovary, ovotestis, and testis of *M**onopterus albus*, one of the most important breeding freshwater fishes in China. Transcriptome comparison assay suggests that a distinct molecular signature exists in each type of GSC, and that different signaling activities are required for the maintenance of distinct GSCs. Functional analysis shows that fGSCs can successfully colonize and contribute to the germline cell lineage of a host zebrafish gonad after transplantation. Finally, we describe a simple feeder-free method for the isolation and enrichment of GSCs that can contribute to the germline cell lineage of zebrafish embryos and generate the germline chimeras after transplantation.

## 1. Introduction

The swamp eel (*Monopterus albus*) is an evolutionarily important fish widely distributed in tropics and subtropics [1,2]. It is one of the most important breeding fishes in China because of its great market value. In recent years, the wild population of *M. albus* has decreased rapidly due to environmental stress, over-exploitation, and their unique life history [3,4,5,6,7]. Thus, a severe shortage of fish resources has been a key limiting factor in aquaculture. New breeding methods are urgently needed for this important fish to meet the increasing market demand.

GSCs are a group of highly specified and undifferentiated pluripotent cells located in the adult gonad [8]. They share many characteristics with embryonic stem cells (ESCs), including high nucleoplasmic ratio, high activity of alkaline phosphatase, and expression of pluripotency markers such as Nanog and KLF4 [9,10,11,12,13]. In mammals, the existence of male GSCs has been firmly established. However, whether there are female GSCs in the ovary remains elusive [14]. In lower vertebrates, including fish, both female and male GSC lines have been reported, including spermatogonial stem cell (SSCs) lines from medaka (*Oryzias latipes*) and North American catfish (*Ictalurus furcatus*) [10,15] and female germline stem cells (fGSCs) from zebrafish (*Danio rerio*) [16]. Importantly, donor GSCs have been used to produce progeny by transplantation, demonstrating their promising prospect in aquaculture [10,12,15,16].

Unlike zebrafish or medaka, the swamp eel is a protogynous hermaphroditic fish that begins as a functional female and ends as a male after an intermediate intersex stage. The intersex gonad has a mixture of ovary and testis (called ovotestis). This nature of sex differentiation leads to (1) a small brood amount due to the small size of female adults in the earlier stage and (2) difficulty in obtaining a sufficient number of sexually mature males. This has been a bottleneck for swamp eel aquaculture. A recent study has shown that both female and male GSCs exist in the ovotestis of swamp eels [17]. Unfortunately, few studies have focused on the isolation and characterization of GSCs from *M. albus*.

In this study, we successfully isolated GSCs from the ovary, ovotestis, and testis of a swamp eel. Transcriptome comparison assay suggested that a distinct molecular signature exists in each type of GSCs and that different signaling activities are required for the maintenance of distinct GSCs. The isolated GSCs could colonize the gonad of a zebrafish and contribute to the germline of the recipient after transplantation.

## 2. Materials and Methods

### 2.1. Animal Ethics

The swamp eel were purchased from the Baishazhou supermarket (Wuhan, China). All experimental procedures were approved by the Institute of Hydrobiology, Chinese Academy of Sciences, Wuhan, China.

### 2.2. Fish and Sample Preparation

Each animal was decapitated. The gonadal tissues were isolated and rinsed twice with DPBS (Dulbecco’s phosphate-buffered saline) containing 2% P/S (penicillin-streptomycin). Under a dissection microscope, the vessels and adipose tissues were manually removed by forceps. The remaining tissues were then cut into 2–3 mm^2^ pieces by micro scissors, followed by digestion with TrypLE Express enzyme (Gibco, 12604021, Waltham, MA, USA) at 28 °C for half an hour. We added 10% fetal bovine serum (TransGen Biotech, FS101-02, Beijing, China) to stop the enzymatic reaction. The mixtures were centrifuged for 2 min at 1000 rpm. The pellets were washed three times with DPBS containing 2% P/S and were filtered through a 40 μm cell strainer (Jet Biofil, CSS010070, Guangzhou, China). After centrifuge at 1000 rpm for 5 min, the cell pellets were collected and resuspended with 3 mL fresh GSC medium that contained RPM1640, 5% fetal bovine serum (Gibco, 10099141), 5% knockout serum replacement (Gibco, 10828028), 4% swamp eel serum, 1x sodium pyruvate (Sigma, S8636, Burlington, MA, USA), 2 mM L-Glutamine (Sigma, 59202C), 50 μm β-Mercaptoethanol, 15 mM HEPES (Bestbio, BB-19324, Shanghai, China), 15 ng/mL mouse LIF (PeproTech, AF-250-02, Cranbury, NJ, USA), and 20 ng/mL FGF (PeproTech, 100-18B).

### 2.3. Culturing of Germline Stem Cells

The 3 mL cultures described above were transferred to 6-well plates precoated with 0.1% gelatin and were not disturbed for 2 days. The medium was then changed every day. GSC colonies can usually be observed on day 4 or 5. When the cells reached 90% confluence, they were trypsinized into single cells and split at a ratio of 1:3. The morphology of the cells was recorded by a microscope.

### 2.4. Alkaline Phosphatase (AP) Staining

Alkaline phosphatase activity experiments were performed with a Leukocyte Alkaline Phosphatase Kit (Sigma, 86C-1KT) according to the manufacturer’s instructions.

### 2.5. Cell Transplantation

To generate germ-cell-deficient zebrafish recipients, *dnd*-specific antisense morpholino oligonucleotides (*dnd* MO 5′-GCTGGGCATCCATGTCTCCGACCAT-3′) were designed and synthesized by Gene Tools (Philomath, OR, USA). The MOs were dissolved to a final concentration of 1 mM with nuclease-free water and stored at −20 °C. Morpholino oligonucleotides (5′-CCTCTTACCTCAGTTACAATTTATA-3′) were used as controls. The efficiency of the *dnd* MO was tested by co-injecting *dnd* MO with the pEGFP-N1 plasmid fused in frame with the MO target sequences. For each 1-cell stage *TG* (*Piwil*:GFP) zebrafish embryo, 4 ng *dnd* MO was injected. Injected embryos were raised and maintained under standard conditions at 28.5 °C. To detect whether zebrafish primordial germ cells were eliminated by knocking down the *dnd* gene, the day 1 living embryos were anesthetized in 168 mg/L tricaine. Anesthetized embryos were mounted in 1.0% low-melt agarose. Fluorescence images were taken by the stereo microscope (Leica M250, Wetzlar, Germany).

The fGSCs of *M. albus* were labeled with fluorescent membrane dye PKH26 (Sigma-Aldrich, MINI26-1KT, Burlington, MA, USA) for 5 min according to the manufacturer’s instructions. The labeling reaction was stopped by adding an equal volume of FBS for 1 min. After labeling, the cells were washed twice with 1 mL M199 (Gibco) and resuspended in 100 uL M199. The resuspended cells were loaded into a microinjection needle. Before transplantation, 7 dpf zebrafish larvae were anaesthetized with 168 mg/L tricaine and mounted on wet tissue paper. The labeled fGSCs were transplanted into the genital ridge of zebrafish recipients using an oil pressure manual microinjection pump (Eppendorf CellTram 4r Oil, Hamburg, Germany) under a stereo microscope (Olympus, Tokyo, Japan).

### 2.6. RT-PCR and Immunofluorescence Analysis of Gonads of Chimeric Fish

The gonads of swamp eels or zebrafish were frozen in liquid nitrogen and stored at –80 °C. The total RNA was extracted using a TRIzol reagent (Invitrogen) and resuspended in nuclease-free water. The quality of the extracted RNA was confirmed by UV spectrophotometer and agarose gel electrophoresis. The total RNA was reverse transcribed into cDNA using ReverTra Ace qPCR RT Master Mix (Toyobo, Japan). For the RT-nested PCR, the cDNA samples were PCR-amplified using specific primers designed for the *dazl* gene. The primers used for the first round of PCR were *dazl*-F1 (5′-AGTATGCCCTCCTTTCTCCT-3′) and *dazl*-R1 (5′-AATTGATGCCTTGATTGTCC -3′). The primers used for the second round of PCR were *dazl*-F2 (5′-CAGAGCACCCTCACCATACC-3′) and *dazl*-R2 (5′-GAATAGCCGAAAGCCCTTAT-3′). The PCR reaction (20 μL) consisted of cDNA or the product of the first-round PCR, 10 uL 2× Taq MasterMix (CWBIO, Taizhou, China), 8 uL ddH2O, and 0.5 uL of forward and reverse primers (10 μmol/mL). *β-actin* was selected as a reference gene. The primer used for the RT-PCR of both the swamp eel and zebrafish *β*-*actin* was *β*-*actin*-F (5′-TGGATGAGGAAATCGCTGCC-3′) and *β*-*actin*-R (5′-TCTTCTCTCTGTTGGCTTTGGG-3′).

The immunofluorescence was performed using anti-Vasa antibodies (GeneTex, GTX128306, Irvine, CA, USA). The Cy5-goat anti-rabbit IgG (BIOSS, bs-0295G-Cy5, Beijing, China) was served as the secondary antibody. Gonads or gonadal tissue sections were fixed in 4% paraformaldehyde (PFA) solution in PBS overnight at 4 °C. The next day, they were dehydrated in 100% methanol for 15 min at room temperature and stored at −20 °C in 100% methanol for at least two hours. The gonads were cut into small pieces under a stereo microscope and permeabilized in acetone (−20 °C) for 8 min, washed with phosphate-buffered saline containing TritronX-100 (PBST), and incubated with the primary antibody overnight at 4 °C. After extensive washing, the samples were incubated with the secondary antibodies overnight at 4 °C. Finally, the gonad samples were stained with DAPI for 10 min. Fluorescence images were captured with a confocal laser scanning microscope (Leica TCS SP5, Wetzlar, Germany).

### 2.7. Reverse Transcription Polymerase Chain Reaction (RT-PCR), Quantitative Real-Time PCR (qRT-PCR), and RNA-Sequencing

The total RNA was extracted either with the total RNA kit (Foregene, RE-03112) or with the TransZol Up RNA kit (TransGen Biotech). One μg RNA was reverse transcribed into cDNA using the TransScript All-in-One First-Strand cDNA Synthesis SuperMix (TransGen Biotech, China, AT341). The cDNAs were used as templates for RT-PCR or qRT-PCR. All primers were listed in Table 1.

For RNA-sequencing, the isolated GSCs were treated with RNA-easy Isolation Reagent (Vazyme, R701-02). RNA-sequencing was performed by BGI Shenzhen Company (Wuhan, China) and was repeated two times. Differentially expressed genes (DEGs) were defined by *p* value < 0.05 and fold change > 1.5.

### 2.8. Hematoxylin–Eosin (HE) Staining

The gonads were isolated and fixed with Bouin’s Fixative Solution (Solarbio, G2331). The H&E staining was performed by Servicebio Technology Company (Wuhan, China). Briefly, the fixed gonads were embedded with paraffin and sectioned at 5 μm. The sections were stained with hematoxylin staining solution after de-paraffinization and rehydration. Next, the sections were rinsed with water, followed by staining with eosin solution. The slides were photographed with a Zeiss microscope (Axio Observer 3, Oberkochen, Germany).

### 2.9. Immunofluorescence Assay

The gonads and culture GSCs were fixed with 4% paraformaldehyde (PFA). For cryosections, the fixed gonads were dehydrated with a graded series of 10%, 20%, or 30% sucrose solution for at least 2 h. The gonads were embedded with Optimal Cutting Compound (OCT, Tissue-Tek) and frozen with dry ice. The samples were then cut into 10 μm cryosections using a cryostat (Thermo) and mounted onto the histological glass slides precoated with Poly-D-Lysine (Sigma, P7886). The cryosections were air-dried and stored in −20 °C. The GSCs or cryosections were washed with PBST (phosphate-buffered saline, 0.1% Triton X-100) three times for 5 min, followed by a blocking step using 5% normal horse serum in PBST. The cells were then incubated with primary antibodies against PH3 (Santa cruz, sc-8656, 1:100), Vasa (DIA-AN, 3008, 1: 200, Wuhan, China), and Nanos2 (1: 200) at 4 °C overnight. After being washed three times with PBST, the cells were incubated with secondary antibodies (1:500 dilution in blocking buffer, Alexa Fluor 488, Life Technologies) at room temperature for 1 h in the dark. The cell nuclei was counter-stained with DAPI (Sigma, D9542, 1:1000). Images were taken with a Leica SP8 laser scanning confocal microscope (Wetzlar, Germany).

### 2.10. Statistical Analysis

Data were shown in the histogram as means ± S.E. The significance of difference in all graphs was assessed using a two-sample equal variance Student’s *t*-test unless specified. *p* value < 0.05 indicates statistical significance.

## 3. Results

### 3.1. Characterization of GSCs in Ovary, Ovotestis, and Testis

H&E staining was performed to examine the gonads of female, intersexual, and male animals (Figure 1A). Three morphologically different gonads, representing the ovary, ovotestis, and testis, were observed. Oocytes of different developmental stages could be readily detected in the ovary of female animals (Figure 1A(a,b), Appendix A), and spermatocytes and spermatids in male gonads were observed according to their morphology (Figure 1A(e,f), Appendix A). In the ovotestis, both testicular- and ovarian-specific tissues were found, with degrading ovarian cells being located at the side of testicular tissues (Figure 1A(c,d), Appendix A).

Several germline stem cell markers such as *nanos2/3, dnd1*, *cdh1**, sall4, stat3, dazl, sox3*, and *ddx4* have been described [18,19,20,21,22,23,24,25,26,27,28,29,30]. The immunofluorescence (IF) staining of Vasa was used to detect the GSCs in gonads. The IF results showed that Vasa-positive germline stem cells were present in all three types of gonads (Figure 1B). To further confirm this, qRT-PCR analysis was performed to examine the expression of GSC-specific and pluripotency marker genes. Pluripotency marker genes such as *klf4* and *nanog* and GSC-specific markers such as *cdh1*, *stat3, dazl*, and *sox3* were abundantly expressed in all three types of gonads, but not in the non-gonadal spleen tissue (Figure 1C and Appendix A).

### 3.2. Isolation and Culture of fGSCs and SSCs

Next, we isolated the GSCs from the gonads, starting with the ovary. Female gonadal tissues were dissociated by mechanical disruption and enzyme treatment and were filtered through a 40 μm cell strainer. The filtered cells were resuspended and plated into 0.1% gelatin precoated plates with the GSC medium. In our GSC medium, the vast majority of cells failed to attach to the plates, remained unaggregated in suspension, and were discarded when changing the medium (Figure 2A, white arrows). The grape-like cells attached quickly to the plates (Figure 2A, red arrows) and after 3–5 days of culturing, they adopted round or oval morphology, displaying a high nucleoplasmic ratio. These cells kept their oval morphology after passages (Appendix A).

We repeated the experiments with the testicular tissues using the same method. The grape-like cells attached to the plates, proliferated, and reached 90% confluency in a week. The cells adopted the fusiform morphology (Figure 2A and Appendix A), which resembled that of medaka SSCs.

To investigate whether the obtained morphologically distinct cells from the ovary and testis were pluripotent, alkaline phosphatase (AP) staining was performed on day 7 cultures. The results showed that both cell types were AP positive (Figure 2B). To investigate whether they were GSCs, qRT-PCR was performed to examine the expression of GSC marker genes such as *ddx4/vasa* and *sox3*, as well as pluripotency markers such as *klf4* and *nanog*. The results showed that these markers were abundantly expressed in the female and male gonadal-derived cells, compared with the non-gonadal spleen cells (Figure 2C). *Gfra1*, a male-related marker [31,32,33], was primarily detected in testis-derived cells; while *foxl2*, a female-related marker [34,35,36,37,38,39], was highly expressed in ovary-derived cells. Importantly, the expression of the Sertoli marker genes *sox9**a* and *wt1* was barely detected, suggesting that there was little contamination of gonadal somatic cells (Appendix A). Finally, IF staining was applied to analyze the expression of Nanos2 and Vasa in day 3 cultures. The results showed that the grape-like cells were positive for both Nanos2 and Vasa (Figure 2D). Based on these data, we tentatively concluded that the grape-like cells isolated from ovary and testis were fGSCs and SSCs, respectively.

### 3.3. Isolation and Culturing of interGSCs

Using the same strategy with the GSC medium, we isolated the GSCs from the ovotestis. After mechanical and enzymatic dissociation of the ovotesticular tissues, the cells were filtered and cultured using our GSC medium. Only a few grape-like cells were observed (Figure 3A). Nevertheless, these cells readily attached to the plates, proliferated, and reached 90% confluency in 7 days, at which time they were positive for AP staining (Figure 3A,B). Interestingly, two morphologically distinct type of cells, similar to fGSCs and SSCs, were observed when cells were passaged (Figure 3C).

qRT-PCR was performed to examine the expression of GSC and the pluripotency marker genes in day 7 cells. They abundantly expressed *cdh1*, *sox3*, and *ddx4*, as well as *nanog* and *klf4*, similar to fGSCs and SSCs (Figure 2C and Figure 3D). Next, IF experiments were performed using Vasa antibodies for day 7 cells. The results showed that nearly 100% of cells were positive for Vasa (Figure 3E). Based on the above data, we concluded that GSCs existed in the ovotestis. We designated the type of GSCs as interGSCs.

To this point, we had isolated three types of GSCs from the ovary, ovotestis, and testis. Next, we investigated whether the GSCs could be maintained for a longer time in our GSC medium in vitro. Unfortunately, the fGSCs and SSCs could only be maintained for up to 30 days when the cells displayed a very low proliferation rate as revealed by the IF staining of PH3, a cell proliferation marker (Appendix A).

### 3.4. Transcriptome Comparison of fGSCs, interGSCs, and SSCs

The isolation of fGSCs, SSCs, and interGSCs allowed us to assess the gene expression signature of different germline stem cells in *M. albus*. RNA sequencing (RNA-seq) was performed for the three types of GSCs. Approximately 5700 highly expressed genes (FPKM > 10) were shared by three types of GSCs, which included known germline stem cell markers such as *cdh1/2*, *stat3*, and *dnd1* (Figure 4A).

The GSC-type specific genes were identified. For instance, female-biased gene *foxl2* was predominantly expressed in fGSCs, while male-biased genes *gfra1* and *dmrta2* were mainly expressed in SSCs, again supporting that they are authentic fGSCs and SSCs. In addition, *foxo1*, *amh*, *amhr2*, *bmp2*, *hsd17bl2*, *fst*, and *lin28b* were found predominantly expressed in fGSCs, while *spata7*, *tex9*, *inhbb*, *dmrta2*, *bcl6*, *gfra1*, and *bmp4* were highly expressed in SSCs. A panel of genes, including *bmp6*, *id2*, *tdrd1*, *spef2*, *maats1*, and *spag6*, were expressed at much higher levels in interGSCs than in SSCs and fGSCs, suggesting that they might have unique roles in the maintenance of interGSCs. A portion of GSC-type specific genes were shown on the heat map (Figure 4B) and were validated by qRT-PCR (Appendix A).

Next, Kyoto Encyclopedia of Genes and Genomes (KEGG) pathway analysis was performed for the up-regulated and down-regulated DEGs among the three types of GSCs. The KEGG results of down-regulated DEGs between fGSCs and interGSCs showed the enrichment of terms such as MAPK, Wnt, and metabolic signaling pathways, while the KEGG results of up-regulated DEGs revealed the enriched terms such as autophagy, apoptosis, Tgf-β, and MAPK signaling pathways (Figure 4C,D). The KEGG analysis of down-regulated DEGs between interGSCs and SSCs showed the enrichment of terms such as FoxO, cell cycle, MAPK, phagesome, and metabolic signaling pathways, while the KEGG analysis of up-regulated DEGs revealed enriched terms such as Tgf-β, Wnt, mTOR, phagesome, autophage, lysosome, and MAPK signaling pathways (Appendix A). The heat map of the selected DEGs of signaling pathways was shown (Figure 4E).

Thus, the results of the transcriptome comparison analysis revealed that each type of GSCs may possess a unique gene expression signature and that different signaling activities might be required for the maintenance of distinct GSCs.

### 3.5. fGSCs’ Ability to Colonize the Recipient Zebrafish and Contribute to the Germline Cell Lineage after Transplantation

Germline transplantation allowed us to functionally identify the isolated GSCs. First, we generated germ-cell-deficient zebrafish that could be used for surrogate reproduction of swamp eels. Morpholino (MO) oligos targeting the start codon region of *dnd* were synthesized and validated using our previously described transgenic line *T**g(**piwil**1:**egfp-UTRnanos3)*. This line (thereafter known as *piwil1: eGFP*) specifically labeled the whole lifetime of germ cells (Figure 5A) [39]. In 24 hpf (hours post-fertilization) *T**g(**piwil**1:*
*eGFP)* embryos or embryos injected with control MO, the GFP-positive PGCs were distributed in the region where the gonads eventually develop, whereas no GFP-positive cells were detected in most (75/87) of the embryos injected with specific *dnd* MO (Figure 5B). Next, fGSC cultures of *M. albus* labeled with PKH26 were transplanted into genital ridge of 7 dpf (days post-fertilization) MO-injected zebrafish larvae (Appendix A). PKH26-labeled fGSCs of *M. albus* were observed in genital ridge of 2 dpt (days post-transplantation) zebrafish recipients (Appendix A). These zebrafish larvae developed into chimeric adult zebrafish with normal morphology.

To determine whether the gonad of chimeric fish contained the donor germ cells, a RT-nested PCR-based mRNA analysis was performed using *M. albus*-specific primers designed for the *dazl* gene (Appendix A). The RT-nested PCR results showed that the *dazl* mRNAs were detectable in all four ovaries of *M. albus* but completely undetectable in zebrafish ovaries after the first round of PCR with *dazl*-F1 and *dazl*-R1 primers or the second round of PCR with *dazl*-F2 and *dazl*-R2 (Appendix A). This indicated the *dazl* mRNA was *M. albus*-specific. In the gonads of 3 mpt (months post-transplantation) chimeric fish, the first round of RT-PCR results showed that *dazl* mRNA was undetectable, whereas in 3/13 gonads of the chimeric fish, *dazl* mRNA was detectable after the second round of RT-PCR with *dazl*-F2 and *dazl*-R2 (Figure 5C).

To confirm that the PCR bands amplified by the second round of RT-PCR were eel-specific, the amplified bands were cut and sequenced. Multiple sequence alignment results showed that the amplified bands had high nucleotide identities with the *dazl* gene of *M. albus* (Figure 5D). To identify the germ cells of the *M. albus* in the gonads of chimeric zebrafish, immunofluorescence analysis was conducted with a Vasa antibody. The immunofluorescence results showed that a small number of *M. albus* germ cells colonized and survived in the gonads of 3 mpt chimeric zebrafish (Figure 5E). Together, these results demonstrated that fGSC cultures derived from *M. albus* possess the ability to contribute to the germline cell lineage of zebrafish host embryos to generate chimeric gonads.

## 4. Discussion

In our study, an effective method was established to enrich the GSCs from the adult ovary, ovotestis, and testis of swamp eels. The pluripotent stem cell characteristic of GSCs was confirmed through several biological strategies. All three types of GSCs displayed key features of GSCs, including the strong activity of AP staining, typical stem cell morphology, and the ability to contribute to zebrafish germ cell lineage.

To date, the successful isolation and culture of fish GSCs is still challenging, which prevents efficient genetic manipulation and its utilization in aquaculture. In this study, we reported a simple feeder-free method to isolate and enrich GSCs from *M. albus* using our GSC medium. The grape-like GSCs (but not other somatic cells) can attach quickly to the culture plates and proliferate efficiently, which allows the enrichment of GSCs. Importantly, GSCs can also be passaged and cultured for a month in vitro. For instance, our SSCs can be cultured for 8–9 passages (about a month), exhibiting typical stem-cell-like morphology similar to the SG3 line derived from medaka [10,21]. This study represents great progress in the isolation and culturing of GSCs from *M. albus.* Unfortunately, there are problems in the long-term maintenance of GSCs in vitro, and further optimization of the culture technique is needed.

To achieve long-term in vitro culture of GSCs, a better understanding of the GSC biology is a prerequisite. In this study, we were able to study the gene expression profile for each type of GSCs through RNA-seq analysis. We found that each type of GSCs possesses a unique gene expression signature despite expressing certain common GSC marker genes. For instance, *bmp3/6* are highly expressed in interGSCs, while *bmp4/15* are predominantly expressed in SSCs. These membrane-localized markers might be used for isolation or discrimination of fGSCs, interGSCs, and SSCs in *M. albus*. The GSC-specific candidate genes, including *bmps*, need to be validated in the future. The KEGG results of down-regulated DEGs between fGSCs and interGSCs showed the enrichment of terms such as MAPK, Wnt, and metabolic signaling pathways, while the KEGG results of up-regulated DEGs revealed the enriched terms such as Tgf-β, autophagy, apoptosis, and MAPK signaling pathways. The apoptosis and autophagy signaling pathways reflected female germline cell degradation and cell death during sex reversal. Importantly, the transcriptome comparison analysis revealed that different signaling activities might be required for the maintenance of distinct GSCs. The Tgf-β and Wnt signaling pathways were particularly interesting to us. Wnt/β-catenin is a major pathway for ovary development and inhibition of testis formation [40,41,42]. Increasing evidence has shown that the Wnt/β-catenin signaling pathway plays a key role in the ovary development and maintenance of fGSCs [40,41,42]. It has been reported that activation of the Wnt signaling pathway promotes the expression of ovarian genes, and β-catenin has been identified as a key pro-ovarian and anti-testis signaling molecule [40,42,43,44,45]. The activation of β-catenin and E-cadherin (*cdh1*) promotes the proliferation of fGSCs in a mouse ovary [46]. In a mouse testis, overexpression of β-catenin promotes male-to-female sex reversal by activating *f**oxl2* [37]. Mutations of *Wnt4* or *Rspo1* promote female-to-male sex reversal in mice, and stabilization of β-catenin can rescue the phenotype [40,41,42,43]. The BMP signaling pathway is important for stem cell proliferation and differentiation [47,48,49]. Inhibition of BMP signaling results in the premature formation of GSCs [50]. BMP signaling pathway genes *bmp4*, *smad7*, and *Id1* are abundantly expressed in the Sertoli cells of *M. albus*, suggesting that somatic-tissue-derived BMP signaling is important for the maintenance of GSCs [17]. Thus, our data suggested that Wnt and Bmp signaling pathways may play important roles in the maintenance or development of GSCs, which provides key cues for optimizing our GSC medium. In the future, by adding Wnt/BMP agonists or antagonists, we might be able to make a GSC medium that can support the long-term culturing of GSCs in vitro.

Interestingly, we found that two morphologically distinct types of cells similar to fGSCs and SSCs were observed when the GSCs from the ovotestis were cultured. This observation was in line with the recent report showing that common progenitors exist in the intersex gonad [17]. Based on the single cell RNA-sequencing data, five different subtypes of germline stem cells were identified in ovotestis of *M. albus*: two common progenitor cell types, early spermatogonia, late spermatogonia, and fGSCs. The developmental trajectory by pseudotime analysis suggested that the common progenitors exist and have the potential to become fGSCs and SSCs. However, it remains unclear whether the grape-like interGSCs in this study are the common progenitors or the mixture of five subtypes. We will look into this in the future.

The nature of life history or sex development of swamp eels leads to difficulties in obtaining sufficient gametes for aquaculture. In recent years, a baby delivered by a sworn-mother strategy has been utilized, providing a useful alternative to overcome the issue [39,51]. In this study, we injected the GSCs from swamp eels into the zebrafish embryos to investigate whether the gametes can be produced in the recipient. We injected day 7 fGSCs and SSCs into recipient zebrafish gonads that were previously injected with *dnd* MO. Only the fGSCs could successfully colonize the zebrafish gonad and contribute to the germline cell lineages after three months of transplantation. Many recipient zebrafish died on the second day after transplantation, and the rate of transplantation with fGSCs was only 3/13. Nevertheless, the successful transplantation of fGSCs in the zebrafish host demonstrated that our in vitro-cultured fGSCs possessed GSC characteristics. Further optimization is needed for the transplantation of SSCs.

In conclusion, we described a simple feeder-free method for the enrichment and culturing of the GSCs of swamp eels. This is the first time that all three types of GSCs of the important commercial fish were isolated. To support the long-term in vitro culture of each type of GSCs, the optimization of GSC medium and cell culture techniques is required. Modulating Wnt/BMP signaling pathways in our current GSC medium might be a good start. Furthermore, the baby delivered by a sworn-mother strategy also needs to be optimized in order to produce gametes in a zebrafish recipient.

## Figures and Tables

**Figure 1 ijms-23-05861-f001:**
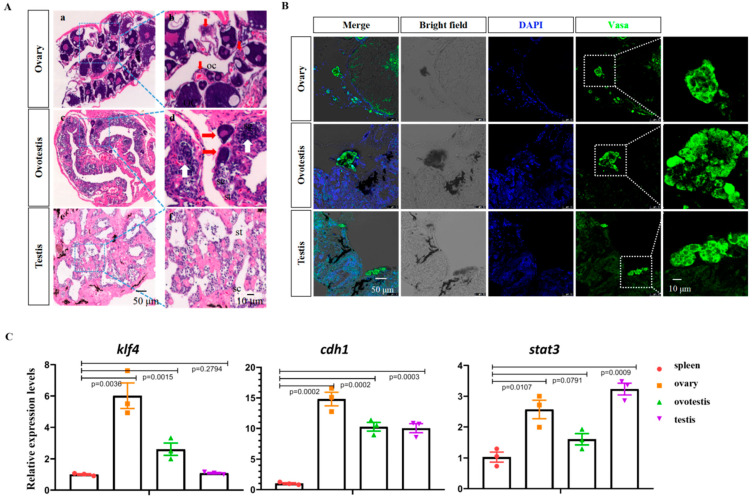
Identification of swamp eel germline stem cells at different stages. (**A**) H&E staining of ovary (**a**,**b**), ovotestis (**c**,**d**), and testis (**e**,**f**) tissues. Red arrows show the oocytes of different stages, and white arrows show the spermatocytes. oc: oocyte; sg: spermatogonia; st: round spermatids. (**B**) Confocal images showing the Vasa-positive cells in the adult ovary, ovotestis, and testis. (**C**) qRT-PCR results showing the expression of pluripotency and GSC marker genes in different types of gonads.

**Figure 2 ijms-23-05861-f002:**
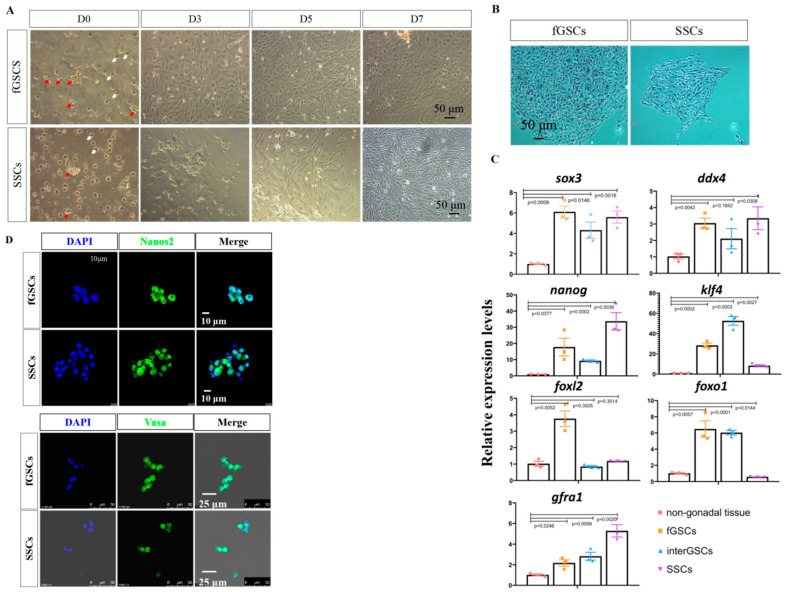
Isolation and characterization of GSCs derived from ovary and testis. (**A**) Cell morphology of GSCs isolated from ovary and testis at the indicated time points. Red arrows show the grape-like GSCs, and the white arrows show the blood cells. (**B**) AP staining images of fGSCs and SSCs. (**C**) qRT-PCR results of pluripotency- and GSC-specific genes in the isolated fGSCs, interGSCs, and SSCs. The spleen was used as the non-gonadal control. (**D**) Immunofluorescence staining results of Nanos2 and Vasa in fGSCs and SSCs.

**Figure 3 ijms-23-05861-f003:**
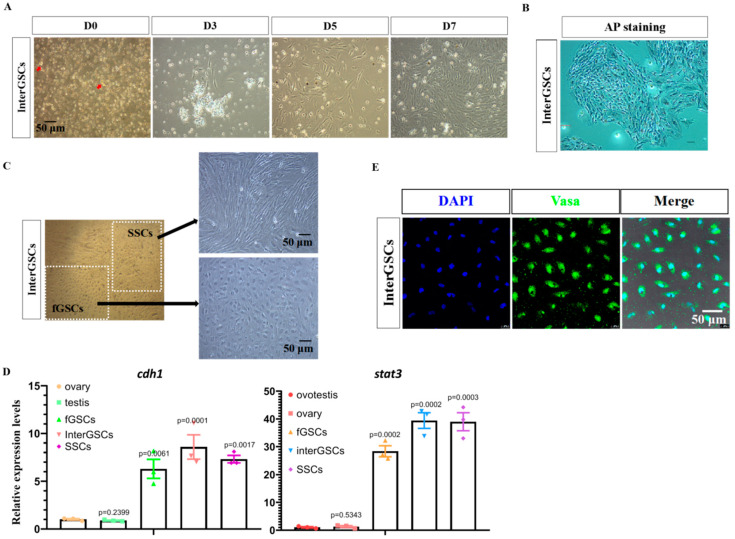
Isolation and characterization of interGSCs derived from ovotestis. (**A**) Cell morphology of GSCs isolated from ovotestis at the indicated time after isolation. Red arrows show the grape-like GSCs. (**B**) Image of AP staining of interGSCs isolated from ovotestis. (**C**) Cell morphology showing the presence of two morphologically distinct cell types. (**D**) qRT-PCR results of *cdh1* and *stat3* in fGSCs, SSCs, and interGSCs, compared with whole ovary, testis, and ovotestis tissues. (**E**) Immunofluorescence images for day 7 interGSCs.

**Figure 4 ijms-23-05861-f004:**
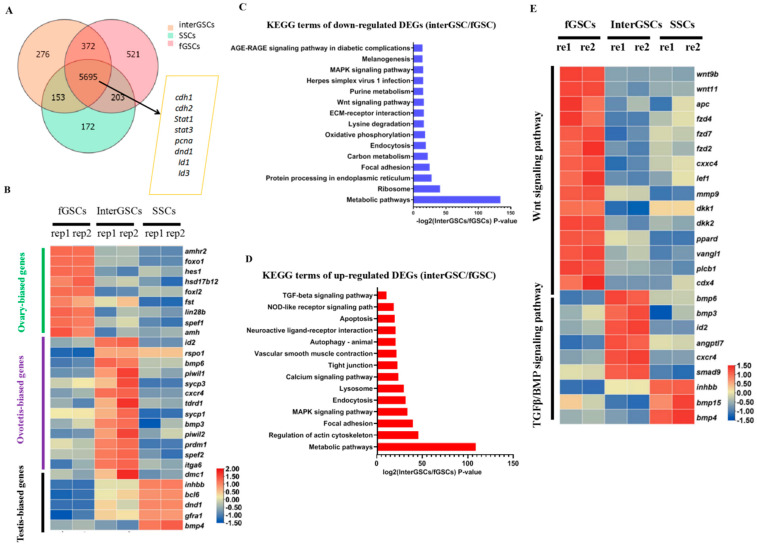
Transcriptome profiling of GSCs derived from ovary, ovotestis, and testis. (**A**) Venn diagram showing the numbers of overlapping genes among fGSCs, interGSCs, and mGSCs, FPKM > 10. (**B**) Heat map showing the expression of the indicated GSC-type specific genes. The color key from blue to red indicates the relative gene expression level from low to high. (**C**) KEGG of down-regulated DEGs in the interGSCs compared with the fGSCs. (**D**) KEGG of up-regulated DEGs in the interGSCs compared with the fGSCs. (**E**) Heat map showing the expression of Wnt- and Tgf-related genes in fGSCs, interGSCs, and SSCs. The color key from blue to red indicates the relative gene expression level, from low to high.

**Figure 5 ijms-23-05861-f005:**
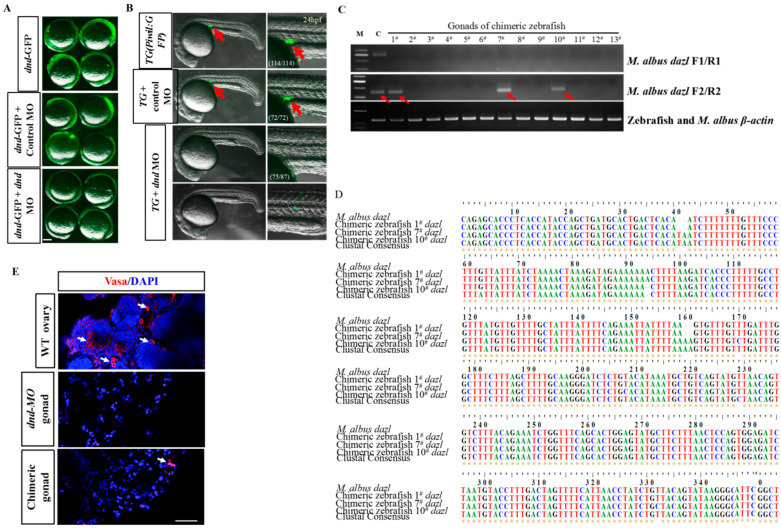
**GSCs contribute to the germline cell lineage of host embryos****to generate zebrafish germline chimeras.** (**A**) The *dnd* MO abolished the expression of dnd-GFP reporter, whereas the control MO did not. (**B**) Images showing GFP signals in embryos injected with control and *dnd* MOs. The GFP-positive PGCs (red arrows) were distributed in the gonadal region of the Tg(*piwil**1:**eGFP*) embryos or embryos injected with control MO. No GFP-positive PGCs were observed in the gonadal region of embryos injected with dnd MO. The frequency of embryos with the indicated phenotypes was shown in the bracket of each group. Lateral view. Scale bars, 500 μm. (**C**) Gel images of RT-PCR products showing the expression of the eel-specific *dazl* gene in gonads of transplanted zebrafish. *β-actin* was used as a reference gene. Red arrows indicate PCR bands amplified by second-round RT-PCR. (**D**) Alignment of the amplified *dazl* sequences from gonads of *M. albus* and gonads of three fGSCs-transplanted zebrafish. (**E**) IF staining of Vasa in ovaries of control-, *dnd* MO-, and *dnd* MO-transplanted with fGSCs zebrafish. White arrows indicate the germ cells. Scale bar = 50 μm.

**Table 1 ijms-23-05861-t001:** The primers used for the RT-PCR analysis.

Genes	Forward (5’-3’)	Reverse (5’-3’)
*elfa*	CGCTGCTGTTTCCTTCGTCC	TTGCGTTCAATCTTCCATCCC
*nanog*	TACGAACTGGGTGTGTGAGC	GGCCAGATAAAAGGCCAGGT
*klf4*	TACCACCATAGCTCCCCACA	GTAGGTTTTCCCACAGCCGA
*sox2*	TCCATGTCCTACTCCCAGCA	CATGTCCCTCAGATCTCCGC
*sox3*	ATGAACGCAGCTTCCACGTA	AGTCCCTGCGGTCTGATAGT
*gfra1*	TGGCTTCTCGTTCCAGATGT	AAGTTGCTCTCCTTACCGCC
*dazl*	GTCCCATCTGGTTGGTCCAG	TCTGGAAATGGGGTGCAACA
*stat3*	ATCCAGTCAGTGGAGCCCTA	AGTCCATGAACACGGAGGGA
*cdh1*	ACTCCCACAAAGAATGACTTCAC	AAAAGGGAATCTTGTGCGGC
*ddx4*	GCTGGATGAAGCTGACAGGAT	AAGGGCCAGAGTTTACCAGT
*dmrt1a*	CTCGCTGGTTAGCTCTGAT	GGAATATGAAACTATCACAAG
*foxl2*	TGACAACAACACGAACAAGGAG	GGCAATGAGAGCGACATAGGA
*sox9a*	GTGAAGAACGGACAGAGCGA	TCGCTGCTGAACTCACCAAT
*foxo1*	CACAGCAGAGCAGCCTCC	CATCTGCCTCCCCAAGATCG
*prdm1*	AGTGTTCAGTTGAAGCCCCC	GGTGAACGAGTTGCAGGGTA
*tdrd1*	TGCAAGCGCTGCAAGAAAAT	ATGCCTGGATGCCTGTTCTC
*smad9*	CCAGAGCACTGGTGTTCCAT	TTGCCTATGTGTCTGCGTGT
*id2*	ACGACTGCTACTCCAAGCTG	GTCCGGCTGTCATCTGTCAT
*bmp6*	CGATCCACACAACCACAGGA	TGTCTGCACAATAGCGTGGT

## Data Availability

Not applicable.

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
