# Peer review of "Isolation and Characterization of Germline Stem Cells in Protogynous Hermaphroditic Monopterus albus"

_ijms, 2022, doi:10.3390/ijms23115861_

Round 1

Reviewer 1 Report

The manuscript entitled 'Isolation and characterization of germline stem cells in protogynous hermaphroditic Monopterus albus' presents interesting results on the characterization of Monopterus albus gonads and transplantation of GSCs into zebrafish.
The text is interestingly written, however I suggest to make some corrections (details below), which will improve the understanding of the text.

e.g. lines 35-36 - please add latin scientific names to common names of fish
line 186 - please correct sentence style
lines 278-282 - in Results we do not discuss results. I suggest to move this fragment to Discussion or combine Results and Discussion sections into one
At the end of the manuscript, please add a few sentences of Conclusions, that would summarize the obtained results, indicate their significance and indicate directions for further research.

Author Response

The manuscript entitled 'Isolation and characterization of germline stem cells in protogynous hermaphroditic Monopterus albus' presents interesting results on the characterization of Monopterus albus gonads and transplantation of GSCs into zebrafish.

The text is interestingly written, however I suggest to make some corrections (details below), which will improve the understanding of the text.

Response: we thank the reviewer for the advice, which greatly helped us with the manuscript.

e.g. lines 35-36 - please add latin scientific names to common names of fish

Response: we have added the latin names, thanks.

line 186 - please correct sentence style

Response: we have revised the sentence.

lines 278-282 - in Results we do not discuss results. I suggest to move this fragment to Discussion or combine Results and Discussion sections into one

Response: we thank the reviewer for the suggestion. We have moved this part of expression into the Discussion section.

At the end of the manuscript, please add a few sentences of Conclusions, that would summarize the obtained results, indicate their significance and indicate directions for further research.

Response: we thank the reviewer for the nice suggestion. We have added the conclusion sentences in which we summarized our results and significance.

Reviewer 2 Report

The manuscript by Sun X. et al describes the isolation and characterization of GSCs from the swamp eel, using gene expression, histology and immunohistochemistry. The GSCs were used for transplantation and could generate chimeric zebrafish.

These results and the overall study must be improved in order to be publishable, since it contains errors and imprecisions that require a careful review.  I have attached the manuscript with notes in specific comments that might help the authors in their revision. A clear description of the objectives of this study is missing, and apparently it reflects in the study design.

The use of anti-Piwi antibodies (but not anti-Cdh1 as acknowledged) is not explained and also why the use of the TG piwil:GFP. The methodology requires clarification and a strong English revision.

  The authors have successfully isolated GSCs from the three types of gonad and could identify a specific pattern of gene expression that is characteristic of these gonads, however the description and presentation of results has many flaws and requires a deep revision. Many genes mentioned in the text are not presented in the figures in results. Some figures are not mentioned in the text (eg. 4A).  Also the use of RT-PCR for making quantification of gene expression is hard to understand why was this option made, specialy when the authors use qPCR in some other instances. 

 The bibliography cited in the text does not correspond to what appears in the link after clicking. This issue must be resolved. As examples there is references P14, 5946 or 7646. 

The discussion is too short and does not cover important issues. Why was the generation of a chimeric zebrafish performed if this is not an endagered species. Others have made simmilar works but none of these studies are mentioned. Also the gene expression data deserved a better and more extended discussion.

 Overall the manuscript needs a careful revision before being sent for a journal for reviewers to evaluate. 

See specific comments in the attached file

Author Response

The manuscript by Sun X. et al describes the isolation and characterization of GSCs from the swamp eel, using gene expression, histology and immunohistochemistry. The GSCs were used for transplantation and could generate chimeric zebrafish.

These results and the overall study must be improved in order to be publishable, since it contains errors and imprecisions that require a careful review. I have attached the manuscript with notes in specific comments that might help the authors in their revision.

Response: we very much thank the reviewer for the detailed comments of our manuscript. We apologize for the errors and imprecisions throughout the text. Helped with the notes by the reviewer, we have revised the manuscript accordingly.

A clear description of the objectives of this study is missing, and apparently it reflects in the study design.

Response: We thank the reviewer for the point. We have revised the introduction part of the manuscript, with a better description of the objectives of the study.

The use of anti-Piwi antibodies (but not anti-Cdh1 as acknowledged) is not explained and also why the use of the TG piwil:GFP. The methodology requires clarification and a strong English revision.

Response: In our lab, we found that Piwi is not expressed in fGSCs and SSCs, but is expressed in the more differentiated cells (not published). So IF of Piwi in Figure 3E was used to show that after 7 days culturing, the GSCs remained in undifferentiated state. Because this observation has not been published (or need to be validated), we decided to leave the Figure 3E out.

The anti-Cdh1 antibodies were not used in this work. So this information in acknowledge has been removed.

In Tg piwil: GFP embryos, the gonadal cells can be labeled with GFP, which allows us to visualize and validate that dnd MO injection can effectively eliminate the gonadal cells of zebrafish recipient embryos (Ye et al., 2019). We have added the information in which it first appears.

We have revised the methods part of the manuscript. Please see the detail.

The authors have successfully isolated GSCs from the three types of gonad and could identify a specific pattern of gene expression that is characteristic of these gonads, however the description and presentation of results has many flaws and requires a deep revision. Many genes mentioned in the text are not presented in the figures in results. Some figures are not mentioned in the text (eg. 4A).

Response: we thank the reviewer for the comments. We apologize for the flaws in the description of gene expression. We have double-checked the text and figures to ensure everything is good.

Also the use of RT-PCR for making quantification of gene expression is hard to understand why was this option made, specialy when the authors use qPCR in some other instances.

Response: we thank the reviewer for the comments. RT-PCR was used to indicate that the genes were expressed in GSCs, which was for qualitative but not for quantitative. Anyway, at the reviewer’s suggestion, we have added the quantitative assay using qRT-PCR.

The bibliography cited in the text does not correspond to what appears in the link after clicking. This issue must be resolved. As examples there is references P14, 5946 or 7646.

Response: we apologize for all this, and have corrected it.

The discussion is too short and does not cover important issues.

Response: we have added information in the discussion part.

Why was the generation of a chimeric zebrafish performed if this is not an endangered species.

Response: The nature of life history or sex development of swamp eel leads to difficulties in the obtaining of gametes for aquaculture. A baby delivered by a sworn-mother strategy has been utilized, providing a useful alternative to overcome the issue (Ye et al., 2019). We injected the GSCs into the zebrafish embryos to investigate whether the gametes can be produced in the recipient.

Others have made similar works but none of these studies are mentioned.

Response: we have added the work by others that are important in the field. I apologize for possible missing some literatures.

Also the gene expression data deserved a better and more extended discussion.

Response: We have added some discussion about the expression data. Please see the third paragraph in the discussion part.

Overall the manuscript needs a careful revision before being sent for a journal for reviewers to evaluate.

Response: we thank the reviewer for the help with the manuscript. We have carefully revised the manuscript according to your advice, many thanks.

Round 2

Reviewer 2 Report

The manuscript format provided must be revised for a reviewer to appreciate it

Author Response

Response to the editor/reviewer

The authors have nicely improved the manuscript but could still perform some minor edits to improve readability and esthetics for reader.

Response: we have revised the manuscript according to the editor. Thanks.

In general figures in panels are on small side and could be enlarged with minor re-organisation (lots of white space in figure panels that could be used to enlarge images/graphics
Response: we have re-organized or adjusted the figures.

Could others provide information on antibodies used (I did not find source info for this)? Pleas also comment on/cite if they have been used in fish previously to detect GSCs. Specificity is always a worry when antibodies are used in new species/tissue

Response: We thank the editor for pointing out the missing information of antibodies and IF information. We have added the“Immunofluorescence Assay”in the methods section (2.9 Immunofluorescence Assay), where the antibody information was added. Please see the detail.   

    For Vasa antibodies, this is a general question regarding Ab specificity in fish studies. We have used the same one that our colleagues have used, please see https://prod--journal.elifesciences.org/articles/66118. Vasa is quite conservative across species, so we think that Vasa antibody is good.

Nanos2 is widely used in zebrafish https://link.springer.com/protocol/10.1007/978-1-4939-4017-2_8. Its suitability in swamp eel has not been published. Nanos2 antibody was kindly provided by our colleague, Dr. Ye Ding, who has tested its suitability for GSCs (personal communications). In this work, the IF data showed that Nanos marks GSCs.

Higher magnification of vasa stained cells as inset in B would be good to actually see cells.

Response: We thank the editor for the comments. Unfortunately, we had no original images at higher magnification. According to the editor,we have zoomed in the Vasa-stained cells, hoping that it will meet your requirement.

What tissue was the q-pcr performed on at start of results? Please see that enough detail is provide for experiments.

Response: the tissues that we used for q-PCR were indicated in the Figure. For instance, Figure 1 C used spleen, ovary, ovotestis, and testis. Maybe I did not get your point, and please state more clearly, thanks.  

Figure 1 C. Colour coding of bars could also be indicated at symbols (circle, square…). Its tricky to identify the tissues just using the black symbols.

Response: We have re-made the figure 1C. Thanks.

Please still check language for typos and errors

Response: I have carefully checked the language, and corrected some typos. Thanks.